# Smokeless tobacco use and oral potentially malignant disorders among people living with HIV (PLHIV) in Pune, India: Implications for oral cancer screening in PLHIV

**Ivan Marbaniang** [1,2]*, **Samir Joshi**[3], **Shashikala Sangle**[4], **Samir Khaire** [5], **Rahul Thakur**[3], **Amol Chavan**[1], **Nikhil Gupte**[1,6], **Vandana Kulkarni** [1], **Prasad Deshpande**[1], **Smita Nimkar**[1], **Vidya Mave**[1,6]

1 Byramjee Jeejeebhoy Government Medical College – Johns Hopkins University Clinical Research Site, Pune, India, 2 Department of Epidemiology, McGill University, Montreal, Québec, Canada, 3 Department of Otorhinolaryngology and Head & Neck Surgery, Byramjee Jeejeebhoy Government Medical College, Pune, India, 4 Department of Medicine, Byramjee Jeejeebhoy Government Medical College, Pune, India, 5 Department of Dentistry, Byramjee Jeejeebhoy Government Medical College, Pune, India, 6 Center for Clinical Global Health Education, Johns Hopkins University, Baltimore, Maryland, United States of America

* ivanmarb@gmail.com

## Abstract

### Introduction

In India, smokeless tobacco (SLT) is a predominant form of tobacco used among people living with HIV (PLHIV). Despite SLT being a risk factor for oral potentially malignant disorders (OPMDs), no prior studies have quantified the association of OPMDs with SLT use among PLHIV. This limits the planning of preventive and control strategies for oral cancer among PLHIV, who are at higher risk for the disease.

### Methods

We enrolled 601 PLHIV and 633 HIV-uninfected individuals in an oral cancer screening study at BJ Government Medical College, Pune, India. Oral cavity images were collected using an m-Health application and reviewed by three clinicians. Participants with two clinician positive diagnoses were deemed to have suspected OPMDs. Prevalence ratios (PRs) were used to quantify the association between suspected OPMDs and SLT use among PLHIV. PRs for current SLT users, across HIV status and use duration were also estimated. Corrected PRs were obtained by modifying the maximum likelihood estimation. Models were adjusted for age, smoking, alcohol use and CD4 counts.

### Results

Of those enrolled, 61% were men, median age was 36 years (IQR: 28–44), and 33% currently use SLT. Proportion of current SLT users was similar across PLHIV and HIV-uninfected groups but use duration for current SLT use was higher among PLHIV(p<0.05). Among PLHIV, current SLT users had a 5-times (95% CI:3.1–7.0) higher prevalence of

disclosed their HIV status except to their HIV health care provider and spouses. Data are available from the Ethics Committee - Byramjee Jeejeebhoy Medical College & Sassoon General Hospitals, Pune, India (contact via email @bjmcecirb@gmail.com) for researchers who meet the criteria for access to confidential data.

**Funding:** This work was supported by amfAR, The Foundation for AIDS Research, with support from the U.S. National Institutes of Health's National Institute of Allergy and Infectious Diseases, the Eunice Kennedy Shriver National Institute of Child Health and Human Development, the National Cancer Institute, the National Institute of Mental Health, the National Institute on Drug Abuse, the National Heart, Lung, and Blood Institute, the National Institute on Alcohol Abuse and Alcoholism, the National Institute of Diabetes and Digestive and Kidney Diseases, and the Fogarty International Center, as part of the International Epidemiology Databases to Evaluate AIDS (IeDEA; U01AI069907) and the NIH-funded Johns Hopkins Baltimore-Washington-India Clinical Trials Unit for NIAID Networks (UM1AI069465). The content and views expressed are those of the authors and does not necessarily represent the official views of any of the governments or institutions mentioned above. The funders had no role in study design, data collection and analysis, decision to publish, or preparation of the manuscript.

**Competing interests:** The authors have declared that no competing interests exist.

suspected OPMDs, compared to non-users. Relative to HIV uninfected individuals with the same SLT use duration, significant associations with suspected OPMDs were seen for PLHIV with<10 use years (PR: 3.5, 95% CI: 1.5–8.1) but not for PLHIV with≥10 use years (PR: 1.3, 95% CI: 0.9–1.8).

## Conclusion

PLHIV that are current SLT users are at high risk of OPMDs and potentially oral cancer. The development of strategies for screening, early detection, and management of OPMDs must be considered for this group.

## Introduction

In 2020, India contributed to more than a third of the global incident cases and deaths related to oral cancer [1]. People living with HIV (PLHIV) have a higher risk of oral cancer [2, 3], and in India this risk is estimated to be as high as 27 times that of the general population [4]. Despite this higher risk, HIV has not been included as a risk factor for oral cancer in the Indian government's operational framework for cancer management [5], and oral cancer preventive measures for PLHIV continue to be lacking.

Oral cancers are often preceded by potentially malignant disorders (OPMDs), which include several premalignant conditions with varying potentials for malignant transformation [6]. The ease of access of the oral cavity without privacy requirements, makes OPMDs readily amenable to detection even in resource limited settings [7]. Moreover, as the estimated average time to malignant transformation is 4–10 years, OPMDs present an important phase of opportunity to introduce interventions that disrupt the natural history of oral cancer [8, 9]. Prevalence estimates of OPMDs for Indian HIV uninfected individuals are among the highest in the world [9, 10]. Although an even higher prevalence of OPMDs is implicated for Indian PLHIV by the higher reported risk of oral cancer [4], no prior studies have quantified this supposition or tested for its validity. To inform the development of oral cancer preventive measures for Indian PLHIV, it is crucial to address this research gap.

According to the 2016–2017 Global Adult Tobacco Survey (GATS-2), the prevalence of adult smokeless tobacco (SLT) use in India is 21.4% (29.6% in men and 12.8% in women) [11]. Due to its cultural acceptability across genders and high prevalence of use, SLT is a predominant risk factor for OPMDs in India [6, 9]. For the Indian general population, SLT use is estimated to be associated with 15-times higher odds of OPMDs [6]. For PLHIV, results from a meta-analysis that used data from the Demographic and Health Survey (DHS) which included 18,224 individuals residing in 28 low- and middle-income countries (LMICs), indicate that the prevalence of SLT use could be 1.3 times higher compared to the general population [12]. However, it is currently not known if the magnitude of the association between OPMDs and SLT use in PLHIV is comparable to the one observed for the general population. This makes it challenging to infer if SLT control strategies should be formulated differently for PLHIV, from those for the general population. SLT use is also only assessed as any (lifetime) or no use in many previous studies [6]. However, as the risk of OPMDs reduces considerably after SLT cessation [13], understanding the association between OPMDs and SLT use specifically among current users while accounting for duration of use, could generate data that is better suited for the development of SLT control strategies that are more resource efficient.

In this study, leveraging an m-Health approach, we sought to compare the prevalence of OPMDs among PLHIV to HIV uninfected individuals; quantify the association of OPMDs with SLT use among PLHIV; and estimate the association between OPMDs and SLT use for current users by use duration, comparing PLHIV to HIV uninfected individuals.

## Methods

### Study setting and design

We used data from an oral cancer screening study conducted at Byramjee Jeejeebhoy Government Medical College and Sassoon General Hospitals (BJGMC-SGH), Pune, India. BJGMC-SGH is a publicly funded tertiary care teaching hospital. It primarily caters to low and lower-middle socioeconomic groups of individuals from the surrounding urban and peri-urban areas of Pune city. Approximately 5000 PLHIV are in active follow-up at the affiliated antiretroviral therapy (ART) center. The annual turnovers of patients at the Otorhinolaryngology and Internal Medicine outpatient clinics are roughly 26,000 and 46,000, respectively.

To calculate the sample sizes for PLHIV and HIV-uninfected participants to be enrolled, we made the following assumptions: 1) The prevalence of OPMDs among HIV-uninfected individuals was assumed to be 10.5%, as reported in a meta-analysis for Asian populations [10]; 2) The prevalence ratio for OPMDs was assumed to be 1.5 among PLHIV compared to HIV-uninfected individuals. Fixing a two-sided confidence interval at 95%, and an enrolment ratio of 1:1 (between HIV-uninfected individuals and PLHIV), we expected to enroll 1320 participants overall (660 PLHIV and 660 HIV-uninfected participants). Under these conditions, we were 80% powered to observe an OPMD prevalence of 15.7% among PLHIV. However, as the study proceeded, due to budgetary constraints, we were unable to meet the originally targeted enrolments. Reducing the power to 77% to detect the same prevalence, the required number of participants was revised to 1226 (613 PLHIV and 613 HIV-uninfected). Power calculations were performed using Epi Info™ StatCalc (Centers for Disease Control—CDC, USA).

PLHIV attending the ART center and HIV uninfected participants from the outpatient clinics of Otorhinolaryngology and Internal Medicine were recruited by convenience sampling. Eligibility criteria included age≥21 years, absence of oral cancer history, and willingness to provide a written informed consent in Marathi, Hindi (locally spoken languages) or English. HIV uninfected individuals had to additionally consent for a rapid HIV test. All PLHIV were on first or second-line ART regimens as recommended by India's national HIV control program. Participants were enrolled into the study between June 2017 and June 2019.

The Institutional Review Board of Johns Hopkins University (JHU) and Ethics Committee of BJGMC-JHU Clinical Research Site approved the project.

### Study procedures

The study procedures described below were first piloted for 20 participants, the data from whom have not been included in subsequent analyses.

**m-Health procedures.** An m-health application jointly developed by investigators at JHU and Boston University as a tool for oral cancer screening was used. The application is designed to populate pre-programmed electronic forms and enable the capture of images. It has been extensively tested in different locations in India [14, 15]. Two trained non-medical health care workers entered participants' information on SLT, smoked tobacco and alcohol use along with sociodemographic, clinical, and sexual history details directly on the application. Eight images of different sites of participants' oral cavities, where OPMDs are commonly diagnosed, were also obtained by health care workers, using an 8-megapixel phone camera. These included: 1) right gingivobuccal mucosa and retromolar trigone; 2) left gingivobuccal mucosa and

retromolar trigone; 3) lower and upper labial mucosa; 4) palate; 5) dorsum of the tongue; 6) floor of mouth and ventral surface of the tongue; 7) right lateral border of the tongue; and 8) left lateral border of the tongue. Data (information and images) were then synced onto a secure cloud-based server.

**Oral Human Papilloma Virus (HPV) procedures.** Participants also provided oral rinse and gargle samples. Detection of oral HPV was performed by using an in-house real-time HPV PCR (GenePath Diagnostics, Pune), and positive samples were sequenced on Illumina MiSeq. Due to technical issues, 230 samples (19 PLHIV, 211 HIV uninfected) were not processed for oral HPV.

**Images review procedures.** Images were accessed and reviewed first by two clinicians (a maxillofacial & oral surgeon and an otorhinolaryngologist), each with at least five years of clinical experience in Head and Neck conditions. Clinicians were provided the age, sex, and oral symptoms (negative and positive) including duration of symptoms (if any) but blinded to the HIV status of participants. No SLT, smoked tobacco or alcohol use details were provided. After reviewing each participant's set of 8 images, depending on whether the clinician considered the images suggestive of having any OPMDs or not, each clinician independently issued a negative or positive 'presumptive OPMD' diagnosis. Clinicians' responses were then reviewed by a study coordinator weekly. If the clinicians disagreed, a senior otorhinolaryngologist with >15 years of clinical experience in Head and Neck oncological surgery served as adjudicator. No prior structured training program to identify suspicious lesions was provided to the any of the clinicians. Participants with two negative presumptive OPMD diagnoses were provided telephonic oral health counselling. Participants with two positive presumptive OPMD diagnoses were asked to come for in-person clinical examinations.

**Follow-up procedures.** Health care workers made up to 5 attempts to call participants with a) two positive presumptive oral HPV diagnoses b) those with a positive oral HPV result. When successfully contacted, participants were instructed to come for an in-person follow-up and their visits were scheduled. For participants that only required telephonic oral health counselling, up to 2 contact attempts were made. All calls were made within two weeks of images review or oral HPV results. Participants successfully linked to clinicians, had relevant clinical examinations performed and were managed according to BGJMC-SGH guidelines.

A scheme of the study procedures is provided in Fig 1.

## Study definitions

**Suspected OPMD (outcome variable).** Classified as positive or negative. A participant with two positive presumptive diagnoses was classified as 'suspected OPMD positive'; a participant with two negative presumptive diagnoses was classified as 'suspected OPMD negative'.

**SLT use (explanatory variable).** Categorized as never, former, and current users, consistent with prior studies [16]. Duration of use for current SLT users was categorized into two groups, <10 and ≥10 years, based on the median use duration among HIV uninfected participants.

## Statistical analysis

Medians and proportions were compared using Wilcoxson rank sum and Fisher's exact tests, respectively. Agreement between clinicians was calculated using the kappa statistic. Modified Poisson regression models with robust estimation of standard errors were fit to estimate prevalence ratios (PRs). First, the PR of suspected OPMDs comparing PLHIV to HIV uninfected individuals was estimated. Second, the analysis was restricted to PLHIV and PRs of suspected OPMDs were estimated across categories of SLT use. Last, PRs of suspected OPMDs were

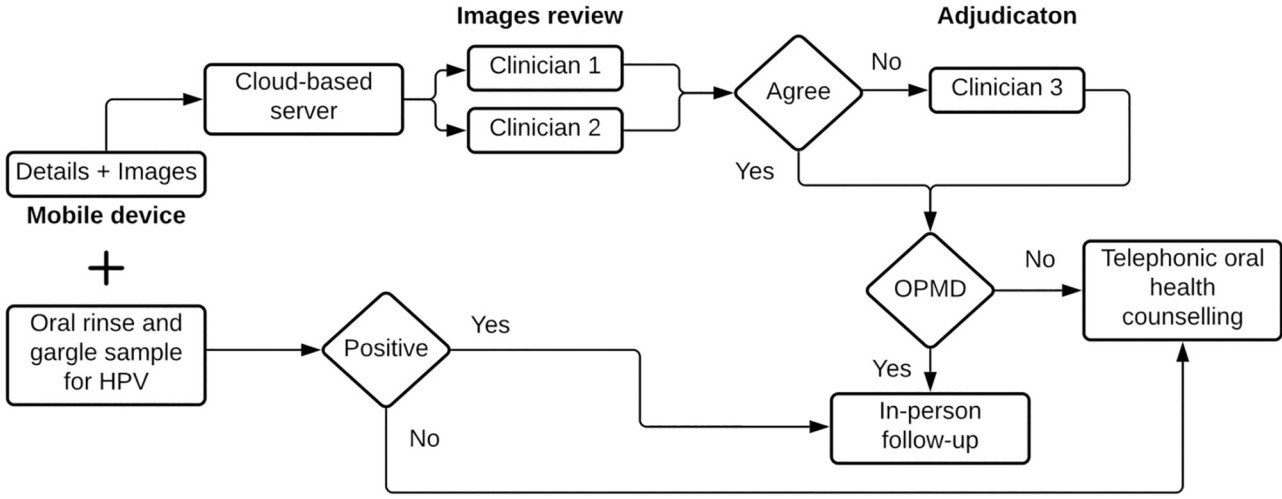

**Fig 1. Scheme of study procedures.**

estimated exclusively for current SLT users by HIV status and use duration. Covariates in multivariable models included variables identified a priori to be associated with OPMDs or that were significantly associated (p<0.05) with suspected OPMDs in univariate models. Covariates included age, smoked tobacco use, alcohol use, and time-updated CD4 counts (only for models restricted to PLHIV). To prevent extrapolation by the model, sex was excluded as a covariate as there were ≤5 women that currently smoke or use alcohol.

We performed three sensitivity analyses to support our primary analyses. 1) We found that the kappa statistic between clinicians ranged between 0.16 to 0.33, indicating poor to slight agreement. Given this low inter-clinician agreement in outcome determination, we posited that the misclassification of the outcome variable would be high (i.e., participants who should have been classified as suspected OPMDs were not and vice versa). Also, only 46% of the participants complied with follow-up procedures. Therefore, we were further unable to validate images review diagnoses with in-person clinical findings. We considered the low compliance of participants with follow-up procedures to be of greater concern than the low agreement between clinicians. Put another way, even if the inter-clinician reliability would have been high (i.e., kappa statistic approaching 1), estimates of prevalence and PRs based exclusively on suspected OPMDs would be less relevant clinically, unless these were validated by in-person examinations. Therefore, we viewed our distribution of the outcome variable (suspected OPMDs) obtained under low inter-clinician reliability as arising from a dataset where the outcome is misclassified relative to in-person clinical examinations (i.e., participants that would have been deemed to have OPMD on clinical examination were not and vice versa). To address this outcome misclassification, we carried out the following procedures: a) validity parameters reported in literature comparing remote (images) to in-person diagnosis of oral conditions were identified [16–18]; b) these parameters were then used to generate 'corrected prevalence' estimates for OPMDs i.e., prevalence estimates for clinical OPMDs in our population, using the formulae specified by Lash et al. [19]. We excluded validity parameters that generated negative estimates; c) validity parameters thus identified were also used to modify the maximum likelihood estimation to obtain 'corrected' odds ratios, adjusted for covariates, as described by Neuhaus, Lyles and Lin, and Lyles et al. [20, 21]. The adjusted odds ratios were then transformed to PRs. Procedures b) and c) were performed assuming non-differential misclassification error of the outcome variable as validity parameters generated external to the

study were used, and clinicians made images review diagnoses independent of SLT use and HIV status knowledge. The range of corrected estimates generated through this sensitivity analysis could thus be more clinically meaningful, as they allowed us to infer what the prevalence and PRs could have been under high participant follow-up conditions. Further, conventionally non-differential misclassification of the outcome has mostly been qualitatively described to bias associations towards the null [22]. Estimation of covariate adjusted corrected associations for non-differential misclassification of the outcome variable is also uncommon [21]. Through this sensitivity analysis, we quantitatively estimated corrected adjusted PRs. For a detailed description of the methodology used to obtain corrected prevalence and PRs, please refer to the supplementary files. 2) Since missing data for oral HPV status (a proposed covariate) was high, PRs were compared with multivariable models in which imputed HPV status values estimated using chained equations were included. Results from models excluding oral HPV status as a covariate were treated as primary findings if estimates from the imputed models were comparable. 3) We compared estimates between models in which sex was excluded as a covariate to those in which it was included.

All analyses were performed using Stata 16.1.

## Results

### Study population

A total of 3112 potential participants were approached (1772 PLHIV, 1340 HIV uninfected individuals). Of these 1234 consented to be enrolled, including 601 PLHIV and 633 HIV uninfected individuals. Males constituted 61% of the study population, and 66% were ≤40 years of age. Thirty-nine percent had ever used SLT, of which 33% were current users. Among current users, the median duration of SLT use was 12 years (IQR: 5–20). The proportions of current alcohol and smoked tobacco users were 26% and 11%, respectively. Oral HPV was detected in 6% (3% oncogenic HPV), and 11% reported ever having performed oral sex. Median time-updated CD4 counts for PLHIV were 508 cells/mm$^3$ (IQR:347–700).

The proportion of current SLT users was similar between PLHIV and HIV uninfected groups (33%). However, SLT use duration among current users was significantly higher among PLHIV than HIV uninfected individuals (PLHIV median SLT use duration:15 years, IQR: 10–20; versus HIV-uninfected SLT median use duration: 10 years, IQR: 4–15, p<0.05) (Table 1).

### Prevalence of suspected and corrected OPMDs

Clinicians reviewed a total of 9,872 images, of which 98% were deemed of good quality to make a diagnosis, and the remaining 2% sufficient to reach a diagnosis. The prevalence of suspected OPMDs was 15% (n = 186) for the entire study population, 19% (n = 117) for PLHIV and 11% (n = 69) for the HIV uninfected group. Depending on the extent of suspected OPMDs misclassification, corrected prevalence estimates ranged between 11.6%–37.0% for the entire study population, 16.3%–48.7% for PLHIV and 6.8%–24.5% for the HIV uninfected group (Fig 2; S1 Fig in S1 File).

### Associations of suspected and corrected OPMDs with HIV status and SLT use

In the unadjusted model, relative to HIV uninfected individuals, the prevalence of suspected OPMDs among PLHIV was 1.79 (95% CI: 1.36–2.35) times higher. When adjusted for covariates, the association remained statistically significant, albeit reduced in magnitude (PR:1.47,

**Table 1. Demographic and clinical characteristics of participants recruited in an mHealth-based oral cancer screening study in Pune, India.**

| | Overall N (%) / Median (IQR) | PLHIV N (%) / Median (IQR) | HIV uninfected N (%) / Median (IQR) |
|---|---|---|---|
| **N (%)** | **1234** | **601** | **633** |
| **Suspected OPMDs** | | | |
| No | 1048 (85) | 484 (81) | 564 (89) |
| Yes | 186 (15) | 117 (19) | 69 (11) |
| **Human Papilloma Virus[†]** | | | |
| Negative | 945 (94) | 536 (92) | 409 (97) |
| Positive | 59 (6) | 46 (8) | 13 (3) |
| **Age (years) [†]** | | | |
| ≤ 30 | 399 (32) | 94 (16) | 305 (48) |
| 31–40 | 414 (34) | 226 (38) | 188 (30) |
| 41–50 | 312 (25) | 219 (36) | 93 (15) |
| > 50 | 109 (9) | 62 (10) | 47 (7) |
| **Sex[†]** | | | |
| Male | 753 (61) | 320 (53) | 433 (68) |
| Female | 481 (39) | 281 (47) | 200 (32) |
| **Smoked tobacco use[†]** | | | |
| Never | 980 (79) | 491 (81) | 489 (77) |
| Former | 122 (10) | 67 (11) | 55 (9) |
| Current | 130 (11) | 42 (7) | 88 (14) |
| **Number of cigarettes/bidis per day** | | | |
| (current tobacco smoker) | 2 (1–3) | 2 (1–4) | 2 (1–3) |
| **Duration of smoked tobacco use[†]** | | | |
| (years) (current smoked tobacco users) | 5 (2–10) | 9 (5–12) | 4 (2–8) |
| **Smokeless tobacco use[†]** | | | |
| Never | 751 (61) | 353 (59) | 398 (63) |
| Former | 70 (6) | 47 (8) | 23 (4) |
| Current | 413 (33) | 201 (33) | 212 (33) |
| **Duration of smokeless tobacco use[†]** | | | |
| (years) (current smokeless tobacco users) | 12 (5–20) | 15 (10–20) | 10 (4–15) |
| **Alcohol use[†]** | | | |
| Never | 766 (62) | 390 (65) | 376 (59) |
| Former | 144 (12) | 88 (15) | 56 (9) |
| Current | 324 (26) | 123 (20) | 201 (32) |
| **Duration of alcohol use[†]** | | | |
| (years) (current alcohol users) | 7 (4–15) | 10 (5–15) | 5 (3–10) |
| **Multiple sexual partners[†]** | | | |
| No | 957 (78) | 445 (74) | 512 (81) |
| Yes | 276 (22) | 155 (26) | 121 (19) |
| **Oral sex** | | | |
| No | 1093 (89) | 538 (90) | 555 (88) |
| Yes | 141 (11) | 63 (10) | 78 (12) |
| **HIV diagnosis duration (years)** | | | |
| < 5 | - | 215 (36) | - |
| 5–10 | | 191 (32) | |
| > 10 | | 195 (32) | |
| **Median first CD4 counts (cells/mm$^3$)** | - | 228 (120–464) | - |

(*Continued*)

**Table 1.** (Continued)

| | Overall N (%) / Median (IQR) | PLHIV N (%) / Median (IQR) | HIV uninfected N (%) / Median (IQR) |
|---|---|---|---|
| **N (%)** | 1234 | 601 | 633 |
| **Median time-updated CD4 counts** (cells/mm$^3$) | - | 508 (347–700) | - |

**Smokeless tobacco use**: use of tobacco forms that are not burnt. Locally available forms are khaini (tobacco + slaked lime); gutka (tobacco + areca nut + slaked lime); mishri (roasted powdered tobacco); paan (tobacco + areca nut + slaked lime + condiments, wrapped in a betel leaf); paan masala; snuff.

**Human Papilloma Virus types included**: HPV 16,18,31,33,35,39,45,51,52,56,58,59.

**Multiple sexual partners** ≥1 lifetime sexual partner; **Oral sex**: performed oral sex in their lifetime.

[†] p-value <0.05 between medians or proportions comparing PLHIV and HIV uninfected.

There are 230 missing values for oral HPV, 211 of which are for the HIV uninfected group, missingness for all other variables is <5%, except first CD4 count (missingness 26.3%).

95% CI: 1.11–1.96) (Table 2, Model 1). Corrected adjusted PRs for the association between HIV positive status and OPMDs ranged between 1.38 to 1.64 (95% CI lower limit range: 1.05–1.14, upper limit range 1.83–2.37) (Fig 3; S2a Fig in S1 File).

When we restricted the analysis to PLHIV, former (PR: 2.26, 95% CI:1.14–4.45) and current (PR: 4.57, 95% CI 3.11–6.70) SLT users had a higher prevalence of suspected OPMDs compared to never users, in the unadjusted model. When adjusted for covariates, the association for current (PR: 4.63, 95% CI: 3.06–7.01) but not former (PR: 2.04, 95% CI: 0.96–4.34) SLT users remained statistically significant (Table 2, Model 2). Corrected adjusted PRs for current users ranged between 4.82–8.88 (95% CI lower limit range: 3.01–3.99, upper limit range: 7.69–21.42) (Fig 3; S2b Fig in S1 File).

Among current SLT users, compared to HIV uninfected individuals and stratified by use duration, PLHIV with <10 years of use had a higher adjusted prevalence of suspected OPMDs (PR: 3.46, 95% CI: 1.48–8.09). A non-statistically significant association was seen for PLHIV with ≥10 years of use (PR: 1.28, 95% CI: 0.90–1.81) (Fig 4). The range of corrected adjusted PRs for PLHIV<10 years of use was 3.35–12.37 (95% CI lower limit range: 1.46–2.48, upper limit range: 7.72–61.63), and for PLHIV≥10 years of use was 1.27–1.39 (95% CI lower limit range: 0.76–0.91, upper limit range: 1.76–1.95).

Findings from sensitivity analyses indicate that the addition of sex or imputed values of HPV as covariates did not affect our primary findings. However, former SLT use (PR: 2.20, 95% CI: 1.03–4.69) became significantly associated with suspected OPMDs in the model that additionally adjusted for sex. This finding is most likely an extrapolation by the model as discussed earlier (S1 and S2 Tables and S3 Fig in S1 File).

## Discussion

In this paper, using data from an m-Health based oral cancer screening study, we estimated the prevalence of OPMDs in Indian PLHIV and compared it with the prevalence in HIV uninfected individuals. Additionally, to our knowledge, we quantified the associations between SLT use and OPMDs in PLHIV for the first time and obtained more granular estimates for the associations between current SLT use and OPMDs by use duration and HIV status. We found that PLHIV were 4%–6% more likely to have OPMDs compared to HIV uninfected individuals, consistent with the known relationship between HIV seropositivity and oral cancer risk [2, 3]. Among PLHIV, relative to non-users, the prevalence of OPMDs was 5–9 times higher among current SLT users. Taken together with the higher prevalence of OPMDs observed among PLHIV, our results indicate that PLHIV that are current SLT users could have the

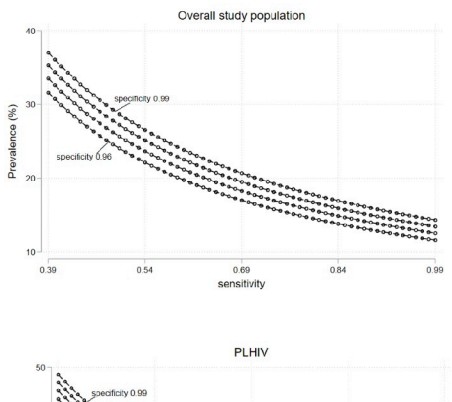

| Prevalence estimates for the overall population (%) | | | |
|---|---|---|---|
| Sp=0.96, Se=0.39 | Sp=0.97, Se=0.39 | Sp=0.98, Se=0.39 | Sp=0.99, Se=0.39 |
| Prevalence= 31.6 | Prevalence= 33.5 | Prevalence= 35.3 | Prevalence= 37.0 |
| Sp=0.96, Se=0.54 | Sp=0.97, Se=0.54 | Sp=0.98, Se=0.54 | Sp=0.99, Se=0.54 |
| Prevalence= 22.1 | Prevalence= 23.7 | Prevalence= 25.1 | Prevalence= 26.6 |
| Sp=0.96, Se=0.69 | Sp=0.97, Se=0.69 | Sp=0.98, Se=0.69 | Sp=0.99, Se=0.69 |
| Prevalence= 17.0 | Prevalence= 18.3 | Prevalence= 19.5 | Prevalence= 20.7 |
| Sp=0.96, Se=0.84 | Sp=0.97, Se=0.84 | Sp=0.98, Se=0.84 | Sp=0.99, Se=0.84 |
| Prevalence= 13.8 | Prevalence=14.9 | Prevalence= 15.9 | Prevalence= 16.9 |
| Sp=0.96, Se=0.99 | Sp=0.97, Se=0.99 | Sp=0.98, Se=0.99 | Sp=0.99, Se=0.99 |
| Prevalence= 11.6 | Prevalence= 12.6 | Prevalence= 13.5 | Prevalence= 14.4 |
| Sp – Specificity, Se – Sensitivity, Prevalence estimates are reported as % | | | |

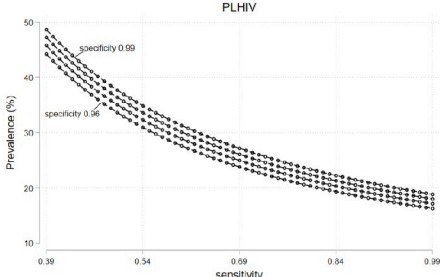

| Prevalence estimates for PLHIV (%) | | | |
|---|---|---|---|
| Sp=0.96, Se=0.39 | Sp=0.97, Se=0.39 | Sp=0.98, Se=0.39 | Sp=0.99, Se=0.39 |
| Prevalence= 44.3 | Prevalence= 45.8 | Prevalence= 47.3 | Prevalence= 48.7 |
| Sp=0.96, Se=0.54 | Sp=0.97, Se=0.54 | Sp=0.98, Se=0.54 | Sp=0.99, Se=0.54 |
| Prevalence= 30.9 | Prevalence= 0.32 | Prevalence= 33.6 | Prevalence= 35.0 |
| Sp=0.96, Se=0.69 | Sp=0.97, Se=0.69 | Sp=0.98, Se=0.69 | Sp=0.99, Se=0.69 |
| Prevalence= 23.8 | Prevalence= 0.25 | Prevalence= 26.1 | Prevalence= 27.2 |
| Sp=0.96, Se=0.84 | Sp=0.97, Se=0.84 | Sp=0.98, Se=0.84 | Sp=0.99, Se=0.84 |
| Prevalence= 19.4 | Prevalence= 20.4 | Prevalence= 21.3 | Prevalence= 22.2 |
| Sp=0.96, Se=0.99 | Sp=0.97, Se=0.99 | Sp=0.98, Se=0.99 | Sp=0.99, Se=0.99 |
| Prevalence= 16.3 | Prevalence= 17.2 | Prevalence= 18.0 | Prevalence= 18.9 |

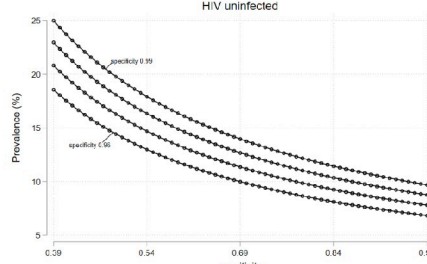

| Prevalence estimates for HIV-uninfected individuals (%) | | | |
|---|---|---|---|
| Sp=0.96, Se=0.39 | Sp=0.97, Se=0.39 | Sp=0.98, Se=0.39 | Sp=0.99, Se=0.39 |
| Prevalence= 18.5 | Prevalence= 20.8 | Prevalence= 22.9 | Prevalence= 24.9 |
| Sp=0.96, Se=0.54 | Sp=0.97, Se=0.54 | Sp=0.98, Se=0.54 | Sp=0.99, Se=0.54 |
| Prevalence= 12.9 | Prevalence= 14.7 | Prevalence= 16.3 | Prevalence= 17.9 |
| Sp=0.96, Se=0.69 | Sp=0.97, Se=0.69 | Sp=0.98, Se=0.69 | Sp=0.99, Se=0.69 |
| Prevalence= 9.9 | Prevalence= 11.3 | Prevalence= 12.7 | Prevalence= 13.9 |
| Sp=0.96, Se=0.84 | Sp=0.97, Se=0.84 | Sp=0.98, Se=0.84 | Sp=0.99, Se=0.84 |
| Prevalence= 8.1 | Prevalence= 9.2 | Prevalence= 10.4 | Prevalence= 11.4 |
| Sp=0.96, Se=0.99 | Sp=0.97, Se=0.99 | Sp=0.98, Se=0.99 | Sp=0.99, Se=0.99 |
| Prevalence= 6.8 | Prevalence= 7.8 | Prevalence= 8.7 | Prevalence= 9.7 |

**Fig 2. Corrected prevalence of OPMDs under different plausible sensitivity and specificity parameters assuming non-differential misclassification of suspected OPMDs (i.e., Sensitivity$_{PLHIV}$ = Sensitivity$_{HIV\ uninfected}$; Specificity$_{PLHIV}$ = Specificity$_{HIV\ uninfected}$).**

highest risk of OPMDs of any group. Our findings highlight the need to plan oral cancer preventive measures for PLHIV, especially for current SLT users.

Cessation of SLT use among PLHIV that are current users would require behavioral modification and as 49% of SLT users in India express no desire to quit [11], primary preventive measures might be challenging to implement. In this context, screening, early detection, and management of OPMDs could be more practicable as a preventive strategy. Preliminary data from Taiwan suggests that screening for OPMDs can be successfully integrated into primary care [23]. A similar approach culturally adapted for India and integrated into routine HIV care could greatly benefit PLHIV that are current SLT users. Oral cancer screening targeted at high-risk groups has been shown to reduce 15-year oral cancer incidence and mortality by 38% and 81%, respectively [24]. Additionally, findings from a recent study indicate that screening targeted at the highest risk group improves resource efficiency without compromising diagnostic sensitivity [25]. Given India's low spending on public health and its overburdened health system [26], our results provide preliminary evidence to support the prioritization of PLHIV that are current SLT users, when oral cancer screening programs for PLHIV are formulated to optimize resource allocation.

**Table 2. Prevalence ratios for suspected oral potentially malignant disorders among participants recruited in an mHealth-based oral cancer screening study in Pune, India.**

| | Suspected OPMDs n (%) | Unadjusted Prevalence Ratio (95% CI) | Adjusted Prevalence Ratio [‡] (95% CI) |
|---|---|---|---|
| **Model 1: Total study population (n = 1234)** | | | |
| **HIV status** | | | |
| Uninfected | 69 (11) | Ref | - |
| Living with HIV | 117 (19) | *1.79 (1.36–2.35)* | *1.47 (1.11–1.96)* |
| **Model 2: People living with HIV (n = 601)[†]** | | | |
| **Smokeless tobacco use** | | | |
| Never | 30 (8) | Ref | Ref |
| Former | 9 (19) | *2.25 (1.14–4.45)* | 2.40 (0.96–4.34) |
| Current | 78 (39) | *4.57 (3.11–6.70)* | *4.63 (3.06–7.01)* |
| **Smoked tobacco use** | | | |
| Never | 89 (18) | Ref | Ref |
| Former | 17 (25) | 1.40 (0.89–2.20) | 0.88 (0.56–1.39) |
| Current | 11 (26) | 1.44 (0.84–2.48) | 1.08 (0.61–1.91) |
| **Alcohol use** | | | |
| Never | 60 (15) | Ref | Ref |
| Former | 22 (25) | *1.63 (1.06–2.50)* | 0.85 (0.54–1.36) |
| Current | 35 (28) | *1.85 (1.28–2.66)* | 0.95 (0.66–1.37) |
| **Age/5 (years)** | - | *1.16 (1.07–1.26)* | *1.10 (1.01–1.21)* |
| **Sex** | | | |
| Male | 72 (22) | Ref | |
| Female | 45 (16) | 0.72 (0.51–1) | - |
| **Human Papilloma Virus** | | | |
| Negative | 106 (20) | Ref | - |
| Positive | 6 (13) | 0.66 (0.31–1.42) | |
| **Multiple sexual partners** | | | |
| No | 79 (18) | Ref | - |
| Yes | 38 (24) | 1.38 (0.98–1.94) | |
| **Oral sex** | | | |
| No | 104 (19) | Ref | - |
| Yes | 13 (21) | 1.07 (0.64–1.79) | |
| **Time-updated CD4/ 50 (cells/ mm$^3$)** | - | 0.99 (0.96–1.03) | 1.01 (0.98–1.04) |
| **HIV diagnosis duration (years)** | - | 0.99 (0.97–1.02) | - |

[†] Percentages in parentheses for Model 2 are derived by dividing the frequency observed for OPMD in that category by the total number of PLHIV in those categories as presented in Table 1.

[‡] Model 1 was adjusted for age, smoked tobacco use, smokeless tobacco use, and alcohol use. The same variables were adjusted for in Model 2 with CD4 as an additional covariate.

Among current SLT users with <10 years of use, PLHIV were found to have a significantly higher prevalence of OPMDs compared to HIV uninfected individuals, consistent with our previous findings. However, we did not observe a significant association between OPMDs and HIV seropositivity when SLT use duration was ≥10 years. We hypothesize that this observation is driven by survival bias. As mentioned earlier, the estimated average time to malignant

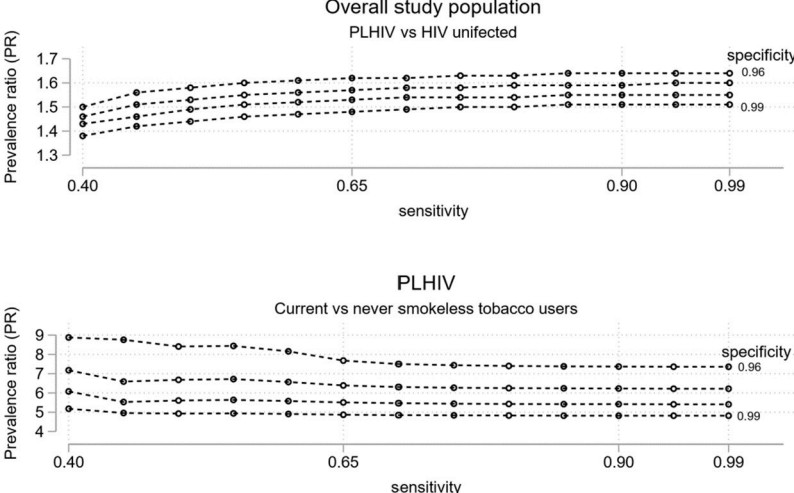

**Fig 3. Corrected adjusted prevalence ratios for OPMDs under different plausible sensitivity and specificity parameters assuming non-differential misclassification of suspected OPMDs.**

transformation of OPMDs is 4–10 years [8, 9]. The paucity of HIV-specific data makes it difficult to surmise if average time would be significantly different in PLHIV. Compared to the general population, oral cancer is reported at younger ages in PLHIV [2], and it is plausible that the average time to malignant transformation is also reduced. Thus, PLHIV who have used SLT≥10 years, could be healthier than an average person living with HIV, given their longevity, contributing to the survival bias. Future longitudinal studies should seek to improve on our findings.

There are several limitations to our study findings. Due to the cross-sectional nature of our study design, risks of OPMDs for SLT use could not be calculated. Nevertheless, considering the extensive evidence that exists for the associations between OPMDs and SLT [6], SLT use

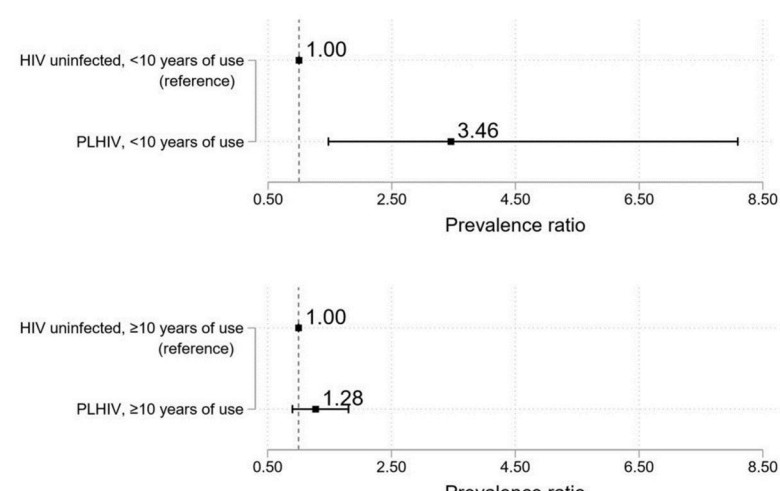

**Fig 4. Adjusted prevalence ratios for suspected OPMDs among current smokeless tobacco users by HIV status and duration of use.**

and HIV [12], and oral cancer and HIV [2–4], our findings have high biological plausibility. Furthermore, by using PRs, we mitigated the overestimation of associations and potentially provide approximations of risk ratios. There were many challenges to study implementation in our setting, namely, low levels of agreement among clinicians and low follow-up compliance of participants, leading to misclassification of the outcome variable, and affecting estimations of prevalence and PRs. Structured clinician trainings to increase inter-clinician agreement and strategies to improve participant retention should be planned in future iterations of similar studies. Nonetheless, we addressed the issues of outcome misclassification by using validity parameters and analytical techniques described in literature [15–19] and generated a range of possible prevalence and PR estimates for our study population. The OPMD prevalence estimates for HIV-uninfected individuals were consistent with estimates reported elsewhere from India [14, 27–30], and the PRs generated were robust to the range of validity parameters used, strengthening our confidence in the results presented. We did not observe a significant association for former SLT users. While this could be related to the frequent resolution of OPMDs with the cessation of SLT use [12], it is possible that we are not adequately powered to detect this relationship. As we did not collect data on the frequency of SLT use, we were unable to assess if the associations observed for SLT use duration are modified by use frequency. Findings from the GATS-2 indicate that 85% of current SLT users in India are daily users and our results are unlikely to vary substantially by use frequency [11]. Smoked tobacco and alcohol use could have been underreported, notably for women, because of their lower social acceptability. We did not collect information on the socioeconomic conditions of participants and are unable to evaluate the extent to which they affect our associations. In India, low socioeconomic position is a determinant of both SLT use and oral cancer mortality [9, 26], and socioeconomic status should be accounted for in future analyses. Lastly, we did not further classify suspected OPMDs into different types, or SLT into different forms. We believe that these details are inconsequential to generate data to plan preventive measures, given that management strategies are similar across different OPMDs and SLT forms [31, 32].

## Conclusion

SLT use is predominantly concentrated in LMICs, and its use is higher among PLHIV compared to HIV uninfected individuals in these countries [12]. However, as a public health problem, SLT use among PLHIV has received considerably less attention than smoking and alcohol use. Viewed through the perspective of health equity, our findings highlight the need to plan oral cancer screening programs for PLHIV in India with a priority for current SLT users, considering that such individuals possibly exist at the intersection of increased oral cancer risk, poverty, and HIV-related social vulnerability.

## Supporting information

**S1 File.**
(DOCX)

**S2 File.**
(DOCX)

## Acknowledgments

The authors would like to thank Dr. Radhika Chigurupati at Boston University and Dr. Robert Bollinger at Johns Hopkins University, co-developers of the m-Health application used in this study. We would like to specially acknowledge Dr. Radhika Chigurupati for all her help in the

initial setting up of the m-Health platform. We thank Dr. Matthew Robinson for providing valuable insights on the first draft of this manuscript. We extend our gratitude to the team at Mueva el Volante especially Markus Aulkemeier for all the technical assistance; Suhasini Surwase and Archana Pawar, the health care workers on this study; Rohini Kamble for her assistance with data entry, and the entire staff at the ART centre. Most of all the authors would like to thank the participants on this study, without whom this work would not be possible.

## Author Contributions

**Conceptualization:** Ivan Marbaniang, Samir Joshi, Shashikala Sangle, Vidya Mave.

**Data curation:** Ivan Marbaniang, Amol Chavan.

**Formal analysis:** Ivan Marbaniang, Amol Chavan.

**Funding acquisition:** Vidya Mave.

**Investigation:** Samir Joshi, Samir Khaire, Rahul Thakur, Prasad Deshpande.

**Methodology:** Ivan Marbaniang, Samir Joshi, Shashikala Sangle, Vandana Kulkarni, Prasad Deshpande, Vidya Mave.

**Project administration:** Ivan Marbaniang, Smita Nimkar.

**Resources:** Vidya Mave.

**Software:** Ivan Marbaniang.

**Supervision:** Samir Joshi, Shashikala Sangle, Nikhil Gupte, Smita Nimkar, Vidya Mave.

**Writing – original draft:** Ivan Marbaniang.

**Writing – review & editing:** Ivan Marbaniang, Samir Joshi, Shashikala Sangle, Samir Khaire, Rahul Thakur, Amol Chavan, Nikhil Gupte, Vandana Kulkarni, Prasad Deshpande, Smita Nimkar, Vidya Mave.

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
