## [Decision Letter · Decision Letter 0]

24 Mar 2022

PONE-D-21-25906Smokeless tobacco use and oral potentially malignant disorders among people living with HIV (PLHIV) in Pune, India: Implications for oral cancer screening in PLHIVPLOS ONE

Dear Dr. Marbaniang,

Thank you for submitting your manuscript to PLOS ONE. After careful consideration, we feel that it has merit but does not fully meet PLOS ONE’s publication criteria as it currently stands. Therefore, we invite you to submit a revised version of the manuscript that addresses the points raised during the review process.

We look forward to receiving your revised manuscript.

Kind regards,

Jitendra Kumar Meena

Academic Editor

PLOS ONE

2. Please note that all PLOS journals ask authors to adhere to our policies for sharing of data and materials: https://journals.plos.org/plosone/s/data-availability. According to PLOS ONE’s Data Availability policy, we require that the minimal dataset underlying results reported in the submission must be made immediately and freely available at the time of publication. As such, please remove any instances of 'unpublished data' or 'data not shown' in your manuscript and replace these with either the relevant data (in the form of additional figures, tables or descriptive text, as appropriate), a citation to where the data can be found, or remove altogether any statements supported by data not presented in the manuscript.

4. Thank you for stating the following in the Funding Section of your manuscript:

“This work was supported by amfAR, The Foundation for AIDS Research, with support from the U.S. National Institutes of Health’s National Institute of Allergy and Infectious Diseases, the Eunice Kennedy Shriver National Institute of Child Health and Human Development, the National Cancer Institute, the National Institute of Mental Health, the National Institute on Drug Abuse, the National Heart, Lung, and Blood Institute, the National Institute on Alcohol Abuse and Alcoholism, the National Institute of Diabetes and Digestive and Kidney Diseases, and the Fogarty International Center, as part of the International Epidemiology Databases to Evaluate AIDS (IeDEA; U01AI069907 as a sub-grant to VM) and the NIH-funded Johns Hopkins Baltimore-Washington-India Clinical Trials Unit for NIAID Networks [UM1AI069465 to VM]. The funders had no role in study design, data collection and analysis, decision to publish, or preparation of the manuscript.”

“This work was supported by amfAR, The Foundation for AIDS Research, with support from the U.S. National Institutes of Health’s National Institute of Allergy and Infectious Diseases, the Eunice Kennedy Shriver National Institute of Child Health and Human Development, the National Cancer Institute, the National Institute of Mental Health, the National Institute on Drug Abuse, the National Heart, Lung, and Blood Institute, the National Institute on Alcohol Abuse and Alcoholism, the National Institute of Diabetes and Digestive and Kidney Diseases, and the Fogarty International Center, as part of the International Epidemiology Databases to Evaluate AIDS (IeDEA; U01AI069907 as a sub-grant to VM) and the NIH-funded Johns Hopkins Baltimore-Washington-India Clinical Trials Unit for NIAID Networks [UM1AI069465]. The funders had no role in study design, data collection and analysis, decision to publish, or preparation of the manuscript.”

6. We note that you have indicated that data from this study are available upon request. PLOS only allows data to be available upon request if there are legal or ethical restrictions on sharing data publicly. For more information on unacceptable data access restrictions, please see http://journals.plos.org/plosone/s/data-availability#loc-unacceptable-data-access-restrictions.

Reviewers' comments:

Reviewer's Responses to Questions

**Comments to the Author**

1. Is the manuscript technically sound, and do the data support the conclusions?

Reviewer #1: Yes

Reviewer #2: Yes

2. Has the statistical analysis been performed appropriately and rigorously? 

Reviewer #1: Yes

Reviewer #2: I Don't Know

3. Have the authors made all data underlying the findings in their manuscript fully available?

Reviewer #1: Yes

Reviewer #2: No

4. Is the manuscript presented in an intelligible fashion and written in standard English?

Reviewer #1: Yes

Reviewer #2: Yes

5. Review Comments to the Author

Reviewer #1: This is a very well-written and thoughtful manuscript. Kudos to the authors for a valuable contribution to the literature. A few suggestions and questions below:

Line 246-250: the low f/u rate (46%), combined with “poor to slight” agreement among clinicians on the OPMD assessment presents a potential challenge in drawing inferences about the outcome variable and its relationship to SLT, HIV status, etc. Perhaps the authors could better explain how the use of validity parameters mitigated this challenge.

Line 173-179: although PRs for OPMD are reported as higher among PLHIV in both adjusted and unadjusted models, the prevalence estimate ranges were reported as overlapping (16.3-48.7 for PLHIV, and 6.8-18.5 for HIV uninfected) when corrected. Estimate range was also very broad for PLHIV in particular. Does this have an impact upon the strength of the conclusions drawn? Are PRs for PLHIV vs. HIV-uninfected not affected by the increased SLT duration noted among the former? If so, this should be mentioned in the Discussion section.

Lines 196-200: while OPV and gender are mentioned, there is no mention of CD4 count. Intuitively, wouldn’t one assume that lower CD4 count was associated with higher rates of OPMD? Perhaps there should be some mention of whether or not this was the case.

Line 268: poverty is mentioned here for the first time (except that BJGMC-SGH caters to low and lower-middle SES patients, line 63-64). This should be mentioned among limitations, as it’s possible that SLT use and/or HIV status in this area is associated with lower SES and/or lower health service utilization, representing a potential confounding contributor to OPMD/oral cancer risk.

Reviewer #2: SLT use among PLHIV is an important public health problem given it’s potential impact on morbidity and mortality of PLHIV. This study aims to “compare the prevalence of OPMDs among PLHIV to HIV uninfected individuals; quantify the association of OPMDs with SLT use among PLHIV; and estimate the association between OPMDs and SLT use for current Users..”

Below are comments and suggested edits

43 Due to its cultural acceptability across genders and high prevalence of use, smokeless

44 tobacco (SLT) is a predominant cause of OPMDs in India [6,9].

Please change consider changing “cause” to risk factor

45………………………………………….. ………………………………..For PLHIV, results

46 from a meta-analysis indicate that the prevalence of SLT use could be twice as high as the general

47 population [11].

Is this just in India or globally? Please clarify (I don’t believe this is true among most PLHIV globally)

It is unclear how the sample size for this study was calculated; what assumptions were made about prevalence of SLT use among PLHIV?

167 to make a diagnosis. The kappa statistic ranged between 0.16 to 0.33, indicating poor to slight

168 agreement among clinicians.

This seems like a problem that casts a big doubt on the main methodology used in this study to assess outcomes of interest including determining the prevalence of suspected OPMDs and link between SLT use and suspected OPMDs among PLHIV. How do you explain the huge discrepancies in results from reading images among the clinicians? In one of the provided references (Birur PN, Sunny SP, Jena S, et al.), the concordance of results between the specialists was almost 100%. I think it is critical to explain the reasons for the poor agreement and how it may have impacted the results.

170 The prevalence of suspected OPMDs was 15% (n=186) for the entire study population,

171 19% (n=117) for PLHIV and 11% (n=69) for the HIV uninfected group.

Again, it is hard to interpret these results without knowing some of the assumptions that went into sample size calculations. For example, what assumptions were made about the agreement between the clinicians in reading images.

176 In the unadjusted model, relative to HIV uninfected individuals, the prevalence of

177 suspected OPMDs among PLHIV was 1.79 (95% CI: 1.36 – 2.35) times higher. When adjusted for covariates, the association remained statistically significant,

In the discussion section the authors did not address the potential impact of HPV on these estimates even though globally HPV prevalence is known to be higher among PLHIV and the association between HPV and cancers (including oral cancers) is well documented.

219 A similar approach culturally adapted for India and integrated into routine HIV care could greatly

220 benefit PLHIV that are current SLT users.

Given the relatively high prevalence of suspected OPMDs among all SLT users (yes higher rates among PLHIV compared with un-infected), wouldn’t it make sense to make this recommendation for all STL users. Yes it is important to prioritize but there are relatively fewer PLHIV SLT users in India compared to the general public who use SLT and it would seem to me that broader public health work around SLT prevention will have a bigger impact compared to limiting work on PLHIV who use STL. This may reduce the risk of coming up with fragmented strategies to address this problem among the different “high risk groups”.

6. PLOS authors have the option to publish the peer review history of their article (what does this mean?). If published, this will include your full peer review and any attached files.

Reviewer #1: **Yes: **Shirish Balachandra

Reviewer #2: No

---

## [Author Response · Author response to Decision Letter 0]

2 Jun 2022

We thank the reviewers for their comments. The comments raised were thoughtful, and they greatly helped us in improving the shortcomings of the previous version of this manuscript. We have tried to address the reviewers’ comments to the best of our ability. If we have been unable to address them, we have added these as limitations of our work. Since the subject of smokeless tobacco (SLT) use and oral potentially malignant disorders (OPMDs) among people living with HIV (PLHIV) has not been very well explored, we hope that this revised version of the manuscript will be considered favorably by the reviewers. 

Please note that the line numbers presented for the reviewers are as they appeared in the original manuscript that had been submitted. Wherever the line numbers are as they appear in the revised manuscript without track changes, this has been explicitly mentioned. 

Reviewer 1

Comment 1: Line 246-250: the low f/u rate (46%), combined with “poor to slight” agreement among clinicians on the OPMD assessment presents a potential challenge in drawing inferences about the outcome variable and its relationship to SLT, HIV status, etc. Perhaps the authors could better explain how the use of validity parameters mitigated this challenge.

Authors’ response: We thank the reviewer for this comment. We agree that the in the originally submitted manuscript the utility of including validity parameters was not explained at length. We have tried our best to clarify how the inclusion of validity parameters was utilized in the analysis and how this is helpful (lines 202 – 229 of the revised manuscript without track changes). 

Briefly, we had two major issues when trying to draw inferences about the associations between HIV status, SLT use and OPMDs. These were:

a) A low level of agreement between reviewing clinicians, as indicated by the low kappa statistic. 

b) High non-compliance with follow-up procedures for in-person clinical examination, as indicated by the low follow-up. 

Both these issues produced misclassification of the outcome variable, albeit in different ways. For example: a) produced misclassification of suspected OPMDs - i.e., participants that would have been classified to have suspected OPMDs were not and vice versa, b) produced misclassification of the outcome variable in relation to in-person clinical examination, i.e., participants that would have been deemed to be negative for OPMDs on in-person clinical examination were classified as OPMD positive and vice versa. Of these two issues, we believe that the non-compliance with follow-up procedures posed a more pressing problem, given that even if the kappa statistic had been 1 (perfect agreement between clinicians), estimates for prevalence and prevalence ratios (PRs) would have been based on images review i.e., suspected OPMDs, whereas to inform clinical practice and policy, we would need to understand how the suspected OPMDs compared against in-person clinical examination diagnoses.

Therefore, we viewed the distribution of the outcome variable obtained under low inter-clinician agreement as one arising from a dataset where there was misclassification of the outcome variable relative to clinical examination. To correct for this misclassification, we used an external set of sensitivity and specificity parameters (derived from studies that had compared images review findings to in-person clinical examination) and obtained a plausible range of estimates for prevalence and PRs. The methodology on how this was done has been described in Part A of the supplementary file. 

Using these validity parameters allowed us to provide corrected prevalence estimates and corrected adjusted PRs that may be more useful to draw out clinical inferences for our study population. 

Comment 2: Line 173-179: although PRs for OPMD are reported as higher among PLHIV in both adjusted and unadjusted models, the prevalence estimate ranges were reported as overlapping (16.3-48.7 for PLHIV, and 6.8-18.5 for HIV uninfected) when corrected. Estimate range was also very broad for PLHIV in particular. Does this have an impact upon the strength of the conclusions drawn? Are PRs for PLHIV vs. HIV-uninfected not affected by the increased SLT duration noted among the former? If so, this should be mentioned in the Discussion section.

Authors’ response: We thank the reviewer for this valuable comment. We recognize that in the originally submitted manuscript it was unclear what the range of the prevalence estimates indicated. We have tried to make this clearer by including Fig 2 in the revised manuscript and shown select numerical prevalence estimates at certain specificity and sensitivity parameters. The range stated for the prevalence estimates is not interpreted as 95% Confidence Intervals would be, thus they cannot be said to overlap, i.e., they do not represent the range of possible prevalence estimates accounting for some level of uncertainty. These are estimates of the prevalence at very specific sensitivity and specificity parameters. 

For example: For PLHIV, a prevalence of 16.3% would only be observed if the sensitivity of the test (comparing diagnosis of OPMD by images review versus in-person clinical examination) is 0.99 and the specificity of the test is 0.96. At these sensitivity and specificity parameters, the only possible prevalence of OPMDs for HIV-uninfected individuals is 6.8%. 

Similarly, for PLHIV, a prevalence of 48.7% is possible only if the sensitivity of the test is 0.39 and the specificity 0.99. At these sensitivity and specificity parameters, the only possible prevalence of OPMDs for HIV-uninfected individuals is 24.9% (this figure has been corrected from the previous figure of 18.5%). If we turn our attention to the Tables below (which are included as part of Figure 2), we will observe that for a given set of sensitivity and specificity parameters, the corrected prevalence estimates between PLHIV and HIV-uninfected individuals do not overlap. 

Prevalence estimates for PLHIV (%) at specific sensitivity and specificity parameters

Sp=0.96, Se=0.39 Sp=0.97, Se=0.39 Sp=0.98, Se=0.39 Sp=0.99, Se=0.39

Prevalence= 44.3 Prevalence= 45.8 Prevalence= 47.3 Prevalence= 48.7

Sp=0.96, Se=0.54 Sp=0.97, Se=0.54 Sp=0.98, Se=0.54 Sp=0.99, Se=0.54

Prevalence= 30.9 Prevalence= 0.32 Prevalence= 33.6 Prevalence= 35.0

Sp=0.96, Se=0.69 Sp=0.97, Se=0.69 Sp=0.98, Se=0.69 Sp=0.99, Se=0.69

Prevalence= 23.8 Prevalence= 0.25 Prevalence= 26.1 Prevalence= 27.2

Sp=0.96, Se=0.84 Sp=0.97, Se=0.84 Sp=0.98, Se=0.84 Sp=0.99, Se=0.84

Prevalence= 19.4 Prevalence= 20.4 Prevalence= 21.3 Prevalence= 22.2

Sp=0.96, Se=0.99 Sp=0.97, Se=0.99 Sp=0.98, Se=0.99 Sp=0.99, Se=0.99

Prevalence= 16.3 Prevalence= 17.2 Prevalence= 18.0 Prevalence= 18.9

Prevalence estimates for HIV-uninfected (%) at specific sensitivity and specificity parameters

Sp=0.96, Se=0.39 Sp=0.97, Se=0.39 Sp=0.98, Se=0.39 Sp=0.99, Se=0.39

Prevalence= 18.5 Prevalence= 20.8 Prevalence= 22.9 Prevalence= 24.9

Sp=0.96, Se=0.54 Sp=0.97, Se=0.54 Sp=0.98, Se=0.54 Sp=0.99, Se=0.54

Prevalence= 12.9 Prevalence= 14.7 Prevalence= 16.3 Prevalence= 17.9

Sp=0.96, Se=0.69 Sp=0.97, Se=0.69 Sp=0.98, Se=0.69 Sp=0.99, Se=0.69

Prevalence= 9.9 Prevalence= 11.3 Prevalence= 12.7 Prevalence= 13.9

Sp=0.96, Se=0.84 Sp=0.97, Se=0.84 Sp=0.98, Se=0.84 Sp=0.99, Se=0.84

Prevalence= 8.1 Prevalence= 9.2 Prevalence= 10.4 Prevalence= 11.4

Sp=0.96, Se=0.99 Sp=0.97, Se=0.99 Sp=0.98, Se=0.99 Sp=0.99, Se=0.99

Prevalence= 6.8 Prevalence= 7.8 Prevalence= 8.7 Prevalence= 9.7

We do not believe that the different prevalence estimates by themselves (depending on the validity parameters) affect the strength of our conclusions drawn, as the corrected adjusted PRs comparing PLHIV and HIV-uninfected individuals remains fairly constant (between 1.38 – 1.64 under different validity parameters, lines 276 - 278 of the revised manuscript without track changes). 

These findings indicate the importance in recognizing that PLHIV are potentially at higher risk of OPMDs than HIV-uninfected individuals, and the need to consider oral cancer screening integration into HIV care in India. Where we believe different prevalence estimates could play a role is in resource allocation. To understand this better, larger studies/surveys where the prevalence of OPMDs among PLHIV are ascertained more rigorously are required. 

Lastly, we addressed the association between SLT use, SLT duration of use, HIV status and OPMDs by stratifying on duration of SLT use (<10 years, ≥10 years) and HIV status in our analysis. These results are presented in Fig 4. As we observe in Fig 4 and have discussed (lines 330 – 340 of the revised manuscript without track changes), we observe a significant positive association (PR: 3.46, 95% CI: 1.48, 8.09) between HIV status and OPMDs for current SLT users when the duration of use is <10 years, but not for when use duration is ≥10 years (PR: 1.28, 95% CI: 0.90, 1.81). If the association between OPMDs and HIV seropositivity was entirely affected by the duration of SLT use, then we would have seen a consistent significant positive relationship even for duration of use ≥10 years. As we discuss (lines 333-340 of the revised manuscript without track changes) we believe the non-significant relationship observed for ≥10 years duration of use is driven partly by selection bias (i.e., this being a cross-sectional analysis, we have selected PLHIV who may be healthier than PLHIV on average, as they have been able to live with HIV, but given their survival have also been able to use SLT longer). We would also like to mention however that even if we had been able to account for selection bias in our analysis, the association with duration of SLT use is a complex issue to discuss. Currently there are no standard methods to assess SLT use duration while also accounting for the quantity of SLT used (like pack-years for cigarette smoking). SLT use duration by itself, we believe is a broad and imperfect measure. Therefore, we have refrained from making too many inferences in this manuscript with respect to the duration of SLT use. 

Comment 3: Lines 196-200: while OPV and gender are mentioned, there is no mention of CD4 count. Intuitively, wouldn’t one assume that lower CD4 count was associated with higher rates of OPMD? Perhaps there should be some mention of whether or not this was the case.

Authors’ response: Thank you for this comment. The objectives of this manuscript (as stated in lines 89 to 92 of the revised manuscript without track changes) are primarily to quantify the associations between SLT use and OPMDs in PLHIV. We had originally mentioned oral HPV and gender in the lines that the reviewer cites as they relate to findings of the sensitivity analyses conducted i.e., how including them as covariates in multivariable models affects the primary associations of interest. We have made it clearer in the revised manuscript (without track changes, lines 298 – 302) that findings related to these two covariates are as they pertain to the sensitivity analyses i.e., their inclusion does not affect the primary associations of interest (estimates are presented in S2 Table of the supplementary files). We agree with the reviewer that CD4 count is an important factor to consider in relation to OPMDs, that is why it is included as a covariate in the multivariable model when the analysis is restricted to PLHIV (Table 2, Model 2). In our analysis we did not find significant associations between CD4 count and OPMDs either in the univariate or multivariable models (Table 2, Model 2) among PLHIV. As CD4 count is not conventionally measured in HIV-uninfected individuals, it was not included as a covariate in models where comparisons with HIV-uninfected individuals were made. 

Comment 4: Line 268: poverty is mentioned here for the first time (except that BJGMC-SGH caters to low and lower-middle SES patients, line 63-64). This should be mentioned among limitations, as it’s possible that SLT use and/or HIV status in this area is associated with lower SES and/or lower health service utilization, representing a potential confounding contributor to OPMD/oral cancer risk.

Authors’ response: We thank the reviewer for this important comment. We have included the lack of socioeconomic information as a limitation in the revised manuscript (lines 363 – 366 of the revised manuscript without track changes).

Reviewer 2 

Comment 5: Line 43 – 44 Due to its cultural acceptability across genders and high prevalence of use, smokeless tobacco (SLT) is a predominant cause of OPMDs in India [6,9]. Please change consider changing “cause” to risk factor.

Authors’ response: We thank the reviewer for this comment. This modification has been made (line 75-76 of the revised manuscript without track changes). 

Comment 6: Line 45 – 47: For PLHIV, results from a meta-analysis indicate that the prevalence of SLT use could be twice as high as the general population.

Is this just in India or globally? Please clarify (I don’t believe this is true among most PLHIV globally)

Authors’ response: Thank you for this comment. We have further qualified the data used in the meta-analysis and corrected the estimate to mention that the prevalence of SLT use could be 1.3 times higher among PLHIV than the general population (line 77 – 80 of the revised manuscript without track changes). 

Briefly, the prevalence of SLT in this meta-analysis was estimated using data from the Demographic and Health Survey (DHS) which included 18,224 individuals residing in 28 low- and middle-income countries (LMICs). Authors found that the relative risk (RR) of SLT use comparing PLHIV to HIV-uninfected individuals was 1·32 [95% CI: 1·03–1·69]; p=0·030 among women and (1·26 [95% CI:1·00–1·58]; p=0·050) among men. 

Comment 7: It is unclear how the sample size for this study was calculated; what assumptions were made about prevalence of SLT use among PLHIV?

Authors’ response: We would like to thank the reviewer for this important comment. It has allowed us to clarify the assumptions made for the sample size of the study population. These have been mentioned in lines 104 -115 of the revised manuscript without track changes. 

• To calculate the sample sizes for PLHIV and HIV-uninfected participants to be enrolled, we first made two assumptions: 

a) The prevalence of OPMDs among HIV-uninfected individuals was assumed to be 10.5%, as reported in a meta-analysis for Asian populations by Mello FW (J Oral Pathol Med. 2018;47(7):633-640) 

b) The prevalence ratio for OPMDs was assumed to be 1.5 among PLHIV compared to HIV-uninfected individuals. 

• We then fixed the two-sided confidence interval at 95%, and an enrolment ratio between HIV-uninfected individuals and PLHIV to 1:1. Under these parameters we expected to enrol 1320 participants overall (660 PLHIV and 660 HIV-uninfected participants) and be 80% powered to detect an OPMD prevalence of 15.7% among PLHIV.

• However, as the study proceeded, we had budgetary constraints and we were unable to meet the original enrolment targets. We therefore revised the power to 77% and the required sample size to 1226 (613 PLHIV and 613 HIV-uninfected participants).

• In the end, we were able to enrol 601 PLHIV and 633 HIV-uninfected individuals. This changed the enrolment ratio to 1.05:1. The other parameters remaining the same, this allowed us to observe an OPMD prevalence of 15.7% among PLHIV powered at 77%.

• For the study population accrued (i.e., 601 PLHIV and 633 HIV-uninfected individuals), assuming the prevalence of SLT use in the general population to be 21.4% (2016-2017 Global Adult Tobacco Survey estimate, line 74 of the revised manuscript without track changes) and the prevalence ratio of SLT use to be 1.32 times higher among PLHIV compared to HIV-uninfected individuals (The Lancet Global Health. 2017;5(6):e578-e592 estimates, line 80 of the revised manuscript without track changes), the study is 80% powered to detect a prevalence of SLT use of 28.4% among PLHIV. 

Comment 8: Line 167 – 168: to make a diagnosis. The kappa statistic ranged between 0.16 to 0.33, indicating poor to slight agreement among clinicians.

This seems like a problem that casts a big doubt on the main methodology used in this study to assess outcomes of interest including determining the prevalence of suspected OPMDs and link between SLT use and suspected OPMDs among PLHIV. How do you explain the huge discrepancies in results from reading images among the clinicians? In one of the provided references (Birur PN, Sunny SP, Jena S, et al.), the concordance of results between the specialists was almost 100%. I think it is critical to explain the reasons for the poor agreement and how it may have impacted the results.

Authors’ response: We thank the reviewer for this comment. We acknowledge that the low kappa statistic posed challenges in determining the prevalence of OPMDs in this study population. However, we believe that the sensitivity analyses we conducted by using validity parameters and the methodology proposed by Lash et al. (Lash T, Fox M, Fink A. Disease Misclassification, Corrections with Sensitivity and Specificity: Nondifferential and Independent Errors. In: Applying Quantitative Bias Analysis to Epidemiologic Data. Springer; 2009:94-96) helped us to provide a plausible range of corrected prevalence estimates for OPMDs. 

The range of corrected prevalence estimates in the HIV-uninfected population (between 6.8% - 24.9%, Fig 2) obtained using the methodology described by Lash et al. corresponds to estimates that have been reported from the country (shown below), which increases our confidence in the results presented.

 Study sample size Prevalence of OPMDs

Birur, Praveen N et al. “Mobile health application for remote oral cancer surveillance.” Journal of the American Dental Association (1939) vol. 146,12 (2015): 886-94 1440 7.4%

Kumar S, Debnath N, Ismail MB, et al. Prevalence and Risk Factors for Oral Potentially Malignant Disorders in Indian Population. Adv Prev Med. 2015;2015:208519. 1241 13.7%

Sivakumar, T T et al. “Prevalence of oral potentially malignant disorders and oral malignant lesions: A population-based study in a municipal town of southern Kerala.” Journal of oral and maxillofacial pathology : JOMFP vol. 22,3 (2018): 413-414. 2368 6.6%

Ramesh, Rohan Michael et al. “Prevalence and determinants of oral potentially malignant lesions using mobile health in a rural block, northeast India.” Tropical doctor vol. 52,1 (2022): 53-60. 2686 26%

Misra, Vatsala et al. “Changing pattern of oral cavity lesions and personal habits over a decade: hospital based record analysis from allahabad.” Indian journal of community medicine : official publication of Indian Association of Preventive & Social Medicine vol. 34,4 (2009): 321-5 753 19.4%

We also acknowledge that while planning the study, we did not foresee the significant disagreement between clinicians. We assumed that given the significant number of years of clinical experience of the clinicians involved in managing Head and Neck conditions, there would have been high agreement, and therefore structured trainings were not warranted. The absence of these trainings, we believe could have significantly contributed to the low kappa statistic. We have included a line about there being no structured trainings for clinicians (lines 160 – 161 in the revised manuscript without track changes) and mentioned the absence of such training as a limitation which must be addressed if similar studies are planned in the future (lines 349 - 350). 

In the Birur PN et at. article the concordance of 100% was calculated differently to how a kappa statistic is calculated (which accounts for the probability that agreement was reached by chance). In the article, only images that were deemed ‘positive’ by a single dental professional were further sent to specialists to be reviewed, and the concordance was based only on positive images. The reviewers were not shown the images of the participants that the single dental professional had deemed negative. In our study, two clinicians simultaneously reviewed the images, and in the event that the first two clinicians disagreed, a third clinician’s opinion was sought. If we calculate the concordance like they did in the Birur PN et.al paper, then we have a concordance of 87%.

Lastly, as we explained in our response to comment 1, there were two main issues when trying to draw inferences about the associations (PRs) between HIV status, SLT use and OPMDs. These were:

a) A high level of disagreement between reviewing clinicians as indicated by the low kappa statistic. 

b) High non-compliance with follow-up procedures for in-person clinical examination. 

Both these issues produced misclassification of the outcome variable, albeit in different ways. For example: a) produced misclassification of suspected OPMDs - i.e., participants that would have been classified to have suspected OPMDs were not and vice versa, b) produced misclassification of the outcome variable in relation to in-person clinical examination, i.e., participants that would have been deemed to be negative for OPMDs on in-person clinical examination were classified as OPMD positive and vice versa. Of these two issues, we believe that the non-compliance with follow-up procedures posed a more pressing problem, given that even if the kappa statistic had been 1 (perfect agreement between clinicians), estimates for prevalence and prevalence ratios (PRs) would have been based on images review i.e., suspected OPMDs, whereas to inform clinical practice and policy, we would need to understand how the suspected OPMDs compared against in-person clinical examination diagnoses.

Therefore, we viewed the distribution of the outcome variable obtained under low inter-clinician agreement as one arising from a dataset where there was misclassification of the outcome variable relative to clinical examination. To correct for this misclassification, we used a wide range of sensitivity and specificity parameters (derived from studies that had compared images review findings to in-person clinical examination) and obtained a plausible range of estimates for prevalence and PRs. The methodology on how this was done has been described in Part A of the supplementary file.

It would have been easier for us to state that the misclassification of the outcome variable (under non-differential misclassification) would bias the PRs towards the null (i.e., make the associations more conservative), but the sensitivity analyses that we performed allowed us to explicitly quantify the range of PRs plausible under a broad range of validity parameters for the study population.

We have expanded on how the low levels of agreement and low follow-up proportions could affect the estimates and how these were handled in the sensitivity analyses (lines 202 - 235 of the revised manuscript without track changes). 

Comment 9: Line 170 – 171: The prevalence of suspected OPMDs was 15% (n=186) for the entire study population, 19% (n=117) for PLHIV and 11% (n=69) for the HIV uninfected group. Again, it is hard to interpret these results without knowing some of the assumptions that went into sample size calculations. For example, what assumptions were made about the agreement between the clinicians in reading images.

Authors’ response: Thank you for this comment. We have mentioned the assumptions that went into the sample size calculations in our response for comment 7. As we mention in comment 8, all clinicians involved in this study are well experienced in managing Head and Neck conditions and we assumed wrongly that there would be high agreement between (at least) two clinicians. Therefore, we did not explicitly account for clinician agreement when we were calculating the sample size requirements. If we had not been constrained by study costs, we would have increased the sample size to account for some part of the misclassification error during the period the study was actively enrolling. However, costs for processing and storing oral HPV samples steeply increased during the financial years of the study, beyond what had been budgeted for, which did not permit us from enrolling more participants.

Comment 10: Line 176 – 177: In the unadjusted model, relative to HIV uninfected individuals, the prevalence of suspected OPMDs among PLHIV was 1.79 (95% CI: 1.36 – 2.35) times higher. When adjusted for covariates, the association remained statistically significant.

In the discussion section the authors did not address the potential impact of HPV on these estimates even though globally HPV prevalence is known to be higher among PLHIV and the association between HPV and cancers (including oral cancers) is well documented.

Authors’ response: We thank the reviewer for this comment. 

• The objectives of this manuscript (as stated in lines 89 to 92 of the revised manuscript without track changes) are primarily to quantify the associations between SLT use and OPMDs in PLHIV. Therefore, to be consistent with these objectives we do not believe that we necessarily need to discuss the association between oral HPV and OPMDs among PLHIV separately. However, we clearly present the association between oral HPV and OPMDs among PLHIV in Table 2 (Model 2). In our analysis, we found that oral HPV is not associated with suspected OPMDs among PLHIV in univariate analysis [Prevalence Ratio 0.66 (95% CI: 0.31 – 1.42)]. 

• As the reviewer has pointed out, we acknowledge that oral HPV is more prevalent among PLHIV and is associated with oral cancer. However, there were 233 and 19 missing oral HPV values among HIV-uninfected individuals and PLHIV, respectively, in our data. Therefore, as part of our sensitivity analyses, recognizing that oral HPV is an important covariate, we first imputed for the missing values before including it in a multivariable model (lines 235-239 of the revised manuscript without track changes). The results of the multivariable model in which oral HPV is included as a covariate are presented in S2 Table (supplementary files). We explicitly mention that the addition of the imputed values of oral HPV as a covariate did not affect our primary findings in the results section of our manuscript (lines 298 – 299 of the revised manuscript without track changes). 

Comment 11: Line 219 - 220 A similar approach culturally adapted for India and integrated into routine HIV care could greatly benefit PLHIV that are current SLT users.

Given the relatively high prevalence of suspected OPMDs among all SLT users (yes higher rates among PLHIV compared with un-infected), wouldn’t it make sense to make this recommendation for all STL users. Yes, it is important to prioritize but there are relatively fewer PLHIV SLT users in India compared to the general public who use SLT and it would seem to me that broader public health work around SLT prevention will have a bigger impact compared to limiting work on PLHIV who use STL. This may reduce the risk of coming up with fragmented strategies to address this problem among the different “high risk groups”.

Authors’ response: We thank the reviewer for this insightful comment. We agree that in an ideal situation in which health services are equitably distributed, recommending an overarching and broad policy to screen all SLT users for OPMDs would equally benefit PLHIV. However, in the current Indian health care context, it is unlikely that recommending such a broad policy (i.e., the top – down approach) would benefit PLHIV, who are potentially at higher risk of OPMDs and oral cancer. 

There are several factors related to HIV care in India, which we believe would not work in favor of a broad policy recommendation. Below we briefly mention two of these factors.

a. Logistical factors: India has the third highest number of PLHIV globally (estimated at 2.3 million). HIV related care is mostly delivered under programmatic conditions (treatment is subsidized by the government) in antiretroviral treatment (ART) centers, which are separate from outpatient clinics used by the general population. HIV medications are provided every 2 weeks to 3 months; CD4 counts/HIV viral RNA tests are performed every 6 months at the ART centers. The integration with other services – for example cancer screening programs is rare in the current HIV care delivery system. At present, PLHIV who wish to avail services like cancer screening need to book another appointment with a separate healthcare provider at a separate clinic. We recommend integrating oral cancer screening programs into routine HIV care in India because it would make access to these services easier (i.e., at the ART center, instead of PLHIV having to visit another health care provider) and can be done regularly (i.e., when PLHIV come for their medications or tests).

b. HIV-related discrimination in health care settings: Studies done across India indicate that HIV-related discrimination in health care settings persist in the country. A few examples are provided below.

• In a study that assessed cervical cancer screening among women living with HIV (WLHIV) in India, it was found that some clinicians involuntarily disclosed the HIV status of the women to others, while other health care providers prioritized HIV-uninfected women or actively discriminated against WLHIV. It was also observed that WLHIV chose to forgo recommended cervical cancer screening and treatment of cervical abnormalities because they were concerned about being discriminated against. Women Health. 2019;59(7):801-814

• In a qualitative study, authors highlight that health care providers often refused to treat PLHIV or referred them elsewhere. BMJ Open. 2019;9(11):e033790

• In a systematic review of 37 papers, authors highlighted discriminatory practices that persist in health care settings throughout India including minimising contact with PLHIV, denying PLHIV assistance during pregnancy, delaying treatment or care, and refusing treatment to PLHIV. SAHARA J. 2011;8(3):138-149

Please note that we are not arguing against oral cancer screening for SLT users in the general population. In fact, as we mention in the introduction (lines 59 - 60 of the revised manuscript without track changes), the government of India has formulated an operational framework for oral cancer management in which it proposes screening all adults>30 years for OPMDs every five years. However, given the persistence of HIV-related discrimination in health care settings in India and considering how HIV care is delivered (point (a) above), it is unlikely that PLHIV will avail oral cancer screening programs intended for the general population. Previous preventive strategies like tuberculosis preventive therapy (TPT) have been successfully integrated into HIV care (while still being available for other tuberculosis high risk groups). We recommend that a similar approach be adopted for oral cancer.

---

## [Editor Report · Decision Letter 1]

20 Jun 2022

Smokeless tobacco use and oral potentially malignant disorders among people living with HIV (PLHIV) in Pune, India: Implications for oral cancer screening in PLHIV

PONE-D-21-25906R1

Dear Dr. Marbaniang,

We’re pleased to inform you that your manuscript has been judged scientifically suitable for publication and will be formally accepted for publication once it meets all outstanding technical requirements.

Kind regards,

Jitendra Kumar Meena

Academic Editor

PLOS ONE

---

## [Editor Report · Acceptance letter]

24 Jun 2022

PONE-D-21-25906R1 

Smokeless tobacco use and oral potentially malignant disorders among people living with HIV (PLHIV) in Pune, India: Implications for oral cancer screening in PLHIV 

Dear Dr. Marbaniang:

I'm pleased to inform you that your manuscript has been deemed suitable for publication in PLOS ONE. Congratulations! Your manuscript is now with our production department. 

Kind regards, 

on behalf of

Dr. Jitendra Kumar Meena 

Academic Editor

PLOS ONE